# What You See is What You Read?
# Improving Text-Image Alignment Evaluation

**Michal Yarom**[G*]   **Yonatan Bitton**[H,G*]   **Soravit Changpinyo**[G]   **Roee Aharoni**[G]

**Jonathan Herzig**[G]   **Oran Lang**[G]   **Eran Ofek**[G]   **Idan Szpektor**[G]

[G]Google Research   [H]The Hebrew University of Jerusalem
michalyarom@google.com, yonatan.bitton@mail.huji.ac.il

## Abstract

Automatically determining whether a text and a corresponding image are semantically aligned is a significant challenge for vision-language models, with applications in generative text-to-image and image-to-text tasks. In this work, we study methods for automatic text-image alignment evaluation. We first introduce SeeTRUE: a comprehensive evaluation set, spanning multiple datasets from both text-to-image and image-to-text generation tasks, with human judgements for whether a given text-image pair is semantically aligned. We then describe two automatic methods to determine alignment: the first involving a pipeline based on question generation and visual question answering models, and the second employing an end-to-end classification approach by finetuning multimodal pretrained models. Both methods surpass prior approaches in various text-image alignment tasks, with significant improvements in challenging cases that involve complex composition or unnatural images. Finally, we demonstrate how our approaches can localize specific misalignments between an image and a given text, and how they can be used to automatically re-rank candidates in text-to-image generation.[1]

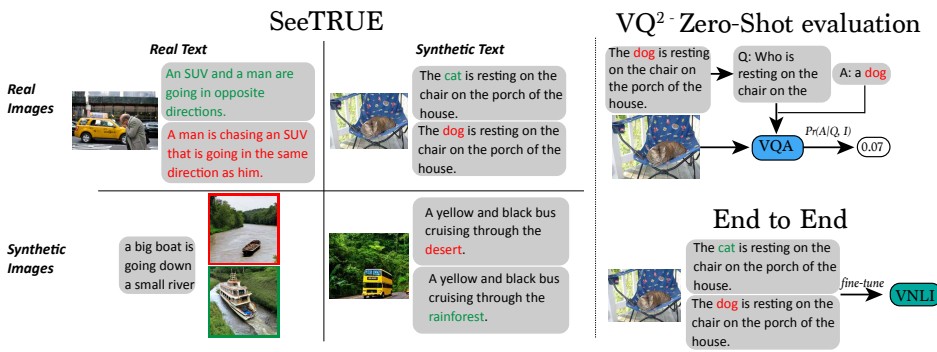

Figure 1: Overview of our approach to text-image alignment evaluation using SeeTRUE. We curate diverse pairs of real and synthetic text and images and use automatic contradiction generation and human evaluation to create a benchmark dataset. We propose two methods for text-image alignment evaluation: VQ$^2$ and VNLI, demonstrated with example pairs.

---

*Equal contribution. Yonatan participated in this work as part of an internship at Google Research.
[1]Data and code are attached to this submission.

37th Conference on Neural Information Processing Systems (NeurIPS 2023).

# 1 Introduction

The recent success and proliferation of multi-modal large language models (LLMs) for text-to-image and image-to-text generation [1–8] make such technology increasingly useful for a wide range of creative applications. However, such models still struggle in generating semantically-aligned image-text pairs; in text-to-image generation, models do not cope with complex specifications [9, 10] or fail to map words in the prompt to visual entities [11, 12]. In image captioning, object hallucination is a long-standing challenge [13] with generated captions still being inferior to human-written ones [14].

Given the above, the task of automatically determining whether a given text-image pair is semantically aligned is highly important, as it is useful both for *evaluating* and for *improving* text-to-image and image-to-text models. However, existing evaluation approaches are still far from ideal; common methods like CLIP [15] or BLIP [6, 7] are based on encoding the image and text as fixed-size embeddings, making it hard to model complex semantics [16]. In addition, while the task is relevant both to text-to-image and image-to-text generation, it is usually studied in silo while considering only one of the applications, thus impeding progress.

In this work, we promote a comprehensive approach to evaluating image-text alignment. We introduce SeeTRUE, a diverse evaluation suite which includes a wide range of image-text pairs with human judgments that determine if the image and text are semantically aligned. SeeTRUE encompasses both real and synthetic images and text, allowing the assessment of text-image alignment models' generalization capabilities across various tasks and 31,855 labeled examples from diverse sources. As part of constructing SeeTRUE, we also introduce a novel method for generating contradicting captions from existing ones by prompting a large language model with tailored instructions.

We present two approaches for automatic image-text alignment evaluation. The first, $VQ^2$, utilizes question generation and visual question answering by generating questions related to the text [17] and ensuring that the correct answer is obtained when asking these questions with the provided image. The second method, Visual Entailment[2] (VNLI), involves directly fine-tuning a large pretrained multimodal model to predict if a given image-text pair is semantically aligned. Both strategies are inspired by recent studies on evaluating factual consistency between two texts [18–21].

We conduct comprehensive experiments on SeeTRUE, demonstrating that both our $VQ^2$ and VNLI methods outperform a wide range of strong baselines, including various versions of CLIP [15], COCA [22], BLIP [6, 7], and OFA [23]. While previous work showed that vision-and-language models tend to exhibit sub-optimal "bag-of-words" behavior [16], the $VQ^2$ method particularly excels on datasets with compositional challenges, achieving state-of-the-art results on the Winoground dataset [24] e.g. by improving the *group score* from 16% to 30.5%. Our methods also demonstrate improved performance when evaluating synthetic images (e.g. on DrawBench [5] and EditBench [25]). Finally, we showcase how $VQ^2$ can identify specific sources of misalignment for a given text-image pair and how our methods can re-rank generated image candidates for a given prompt.

To summarize, our contributions are as follows: (1) We introduce the SeeTRUE benchmark for meta-evaluation of image-text alignment. (2) We introduce a novel method to generate contradicting image captions from given captions with LLMs. (3) We suggest two reference-free metrics for image-text alignment evaluation: $VQ^2$, based on question generation and visual question answering, and VNLI, based on fine-tuning large multimodal language models. (4) We conduct extensive evaluation of the above approaches against strong baselines, demonstrating superior performance over multiple datasets. (5) We release our evaluation suite, models and code to foster future work.

# 2 SeeTRUE: A Comprehensive Text-Image Alignment Benchmark

We begin by introducing SeeTRUE, a diverse benchmark for meta-evaluation of image-text alignment methods, covering the 4-way combinations of real and synthetic text-and-image pairs. It addresses limitations in current benchmarks, which mainly focus on natural images and often lack challenging negative captions. SeeTRUE allows to better assess the generalization abilities of text-image alignment models across various tasks.

Defining how image-text alignment is assessed has a direct impact on the construction of evaluation datasets. As images can display more details than described in their caption or text prompt, we define

---

[2]We use the terms Entailment and Natural Language Inference (NLI) interchangeably.

Table 1: SeeTRUE: a benchmark for image-text alignment encompassing 31,855 real and synthetic image-text pairs from diverse datasets and tasks. An example from each dataset is presented below.

| | Real Text + Real Images | | Real Text + Synthetic Images | | | Synthetic + Real | Synthetic + Synthetic |
|---|---|---|---|---|---|---|---|
| **Dataset Name** | SNLI-VE | Winoground | DrawBench | EditBench | COCO t2i | COCO-Con | PickaPic-Con |
| **# Test Examples** | 17,901 | 1,600 | 1,968 | 3,827 | 2,586 | 1,992 | 1,981 |
| **% Positive / Total** | 33.3% | 50% | 55.7% | 36.9% | 63.6% | 52.7% | 44.1% |
| **Labeled in this work?** | ✗ | ✗ | ✓ | ✓ | ✓ | ✓ | ✓ |
| **Image** | | | | | | | |
| **Text** | the player swings his bat | the heavy oncoming traffic is contrasted with the light outgoing traffic | A blue cup and a green cell phone | a few pink candles and some cream on top of a cake. | A person on a snow board high up in the air. | A giraffe leaned over in a plush field next to some cows | a doctor wearing a white coat in the middle of a street |
| **Human Label** | True | True | False | True | False | False | True |

image-text alignment as the case where all the details described in the text are accurately represented within the image. Inspired by the Textual Entailment task [26] which judges for two pieces of text whether one (the "hypothesis") can be inferred given the other (the "premise"), our definition maps the image to the premise and the text to the hypothesis, resulting in the task of predicting whether the information in the text can be inferred from the given image.

## 2.1 Datasets

We describe the datasets included in our benchmark, with a high-level overview in Table 1.

**Real text and real images.** For pairs of human-written text and real (non-generated) images, we include the SNLI-VE [27] and Winoground [24] datasets. SNLI-VE is a widely adopted VNLI dataset containing an image, a text, and a label of the alignment between the two – entailment, contradiction, or neutral. Winoground is a challenging dataset for compositional understanding, where each example includes two images and two text captions, where the task is to match each text to its corresponding image. The captions only differ in a few words, which should result in distinct visual interpretations. For example, "some plants surrounding a lightbulb" vs. "a lightbulb surrounding some plants".

**Real text and synthetic images.** For datasets that represent text-to-image generation tasks we use EditBench [25] which offers prompts and images generated by various text-to-image models given those prompts, accompanied by alignment ratings. To encourage more diversity in the data, we also create new datasets by generating images using Stable Diffusion models [3] (V1.4 and V2.1) and Imagen [5] by prompting them with COCO [28] captions and text prompts from DrawBench [5], creating the COCO text-to-image ("COCO t2i") and the DrawBench text-to-image datasets.

**Synthetic text and real images.** This category includes a new dataset which we name COCO-Con. COCO-Con is generated using a novel automatic method which we describe in detail in Section 2.3. Specifically, we generate synthetic contradicting captions for COCO images based on their original captions by prompting a large language model, and verify the resulting captions with human raters.

**Synthetic text and synthetic images.** We utilize PickaPic [29], a source of user-generated and ranked synthetic images. We create synthetic captions using BLIP2 [7] and employ our automatic contradiction generation method (Section 2.3) to produce unaligned captions. This category evaluates synthetic text that is generated by image captioning models, e.g. for improving textual image search.

We note that some of the datasets are only used for testing (e.g., Winoground, DrawBench, EditBench) while others include both training and test sets (e.g., SNLI-VE, COCO t2i, COCO-Con, PickaPic-Con). This allows us to investigate different training configurations and their effect on performance.

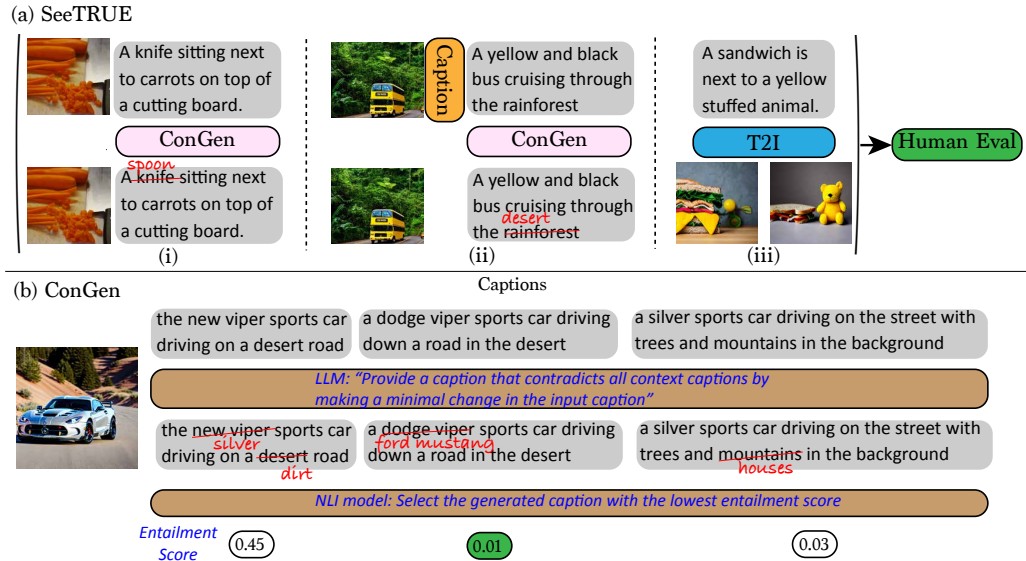

Figure 2: (a) The SeeTRUE generation process. (i) An image-text pair from a dataset is used to generate a contradicting caption using ConGen. (ii) An image (real or synthetic) is passed through a captioning model to generate a caption, which is then passed to ConGen to generate a contradicting caption. (iii) A text-to-image model is applied on captions from the dataset to create multiple image-text pairs. All the resulting examples are evaluated by human raters to create SeeTRUE. (b) The contradiction generation process (ConGen) takes a caption as input and instructs an LLM to generate variants that contradict it. An NLI model is used to select the variant with the lowest entailment score.

## 2.2 Human Annotation and Evaluation

To standardize the labeling scheme across datasets, we follow TRUE [19] and use binary annotations for alignment/misalignment. In datasets with three-way annotations (e.g. Entailment, Contradiction, Neutral) we convert the labels to binary labels by collapsing all non-entailment/non-alignment labels to a single negative label.

Some datasets, such as COCO-Con and PickaPic-Con, start with automatically generated labels, while others lack annotations entirely (e.g. DrawBench). To make sure we have high quality labels we conduct human annotation for all test examples in such datasets. We ask three crowd-workers from Amazon Mechanical Turk (AMT) to evaluate whether a given image-text pair is aligned, by answering the question: "Does the image present all the details described in the text correctly?" with "Yes" or "No". If the answer is "No", the workers are also requested to describe the main misalignment to enhance the annotation quality. While the random chance of agreement is 25%, the annotators reached consensus in 80% of cases. Furthermore, we measured a Fleiss-Kappa [30] score of 0.722, showing a good level of agreement between the annotators. Full annotation details, AMT user interface example, and agreement numbers per dataset can be found in appendix A.3.

The datasets we annotated include DrawBench, COCO t2i, COCO-Con and PickaPic-Con, with statistics presented in Table 1. These datasets vary in their positive/negative distribution, with COCO t2i having the highest percentage of positives (63.6%) and DrawBench having the lowest (36.9%). The agreement with the auto-label is 94% for COCO-Con and 77% for PickaPic-Con. To prevent the inclusion of offensive images, particularly those that are synthetically generated, annotators are asked to mark any images that may be considered offensive and these were discarded.

## 2.3 ConGen: Generating Contradicting Captions by Prompting LLMs

We propose an automatic method for generating unaligned captions from existing, aligned image-and-text pairs, with the goal of creating challenging examples for evaluation and training. Our method is inspired by the concept of contrast sets: given an original example with a corresponding label, we create a minimally perturbed example where the perturbation changes the corresponding

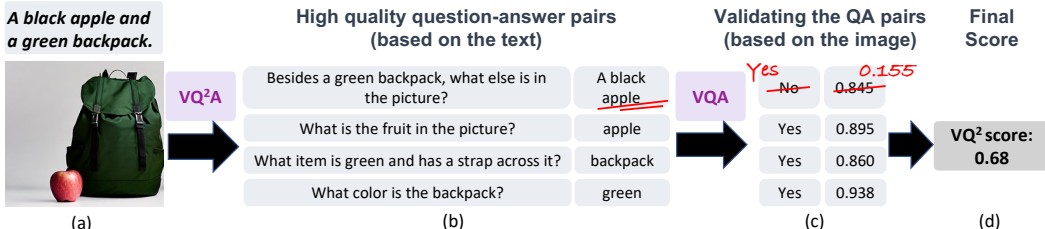

Figure 3: The $VQ^2$ pipeline: (a) given a text and an image, (b) generate question and answer pairs from the text, (c) re-write each pair as a yes-no question and obtain the 'yes' answer probability from the image as an alignment score. (d) Finally, average all alignment pair scores as the final $VQ^2$ score.

label [31–34]. Contrast sets address the issue of supervised models exploiting data artifacts in i.i.d. train/test splits to achieve high test scores, while their performance degrades significantly on samples outside their training distribution.

To create contrast sets for image-text alignment, we go over the text captions from the image-text pairs in the COCO and PickaPic datasets, covering both natural and synthetic images. For each caption we instruct PaLM [35], a large language model, to generate several contradicting captions via few-shot inference with 7 positive and 8 negative examples. For instance, for the caption "*a knife sitting next to carrots on top of a cutting board*", the model replaces the word *knife* with *spoon* (see Fig. 2, left). We then use a Natural Language Inference (NLI) model [18] to score whether the generated caption is indeed contradicting the original, and select the generated caption with the highest contradiction score. Figure 2 illustrates this process. Human annotators verified that the resulting contradicting captions are of high quality, with 94% agreement with human labels in COCO and 77% agreement with human labels in PickaPic (more details in section 2.2).

## 3    Methods

Using our SeeTRUE benchmark, we would like to reassess the performance of multimodal alignment approaches. In this section we introduce two image-text alignment methods. In Section 4 we will compare their performance against established, previously published methods.

### 3.1    $VQ^2$: Zero-Shot Alignment via Question Generation and Visual Question Answering

Inspired by recent work on factual consistency evaluation in text-to-text tasks [36, 18, 37], we propose a zero-shot approach for automatically evaluating image-text alignment based on question generation and question answering. Figure 3 provides an overview of the method. The motivation is to extract question-answer pairs, which capture the important details of the text, and then to validate that they are presented correctly in the image. For a given image-text pair $\{I, T\}$, we first extract a set of candidate answer spans $\{a_j\}_{j=1}^N$ from the given text $T$. Then, we use a question generation (QG) model to generate a question for each answer candidate $q_j = QG(a_j, T)$. Each generated question-answer pair $(q_j, a_j)$ is scored with a question answering (QA) model, and if $QA(q_j, a_j, T)$ returns a low score, we filter out the corresponding pair. This results in a subset of $M$ question-answer pairs $\{(q_j, a_j)\}_{j=1}^M$.

Each generated question-answer pair $(q_j, a_j)$ is then independently validated based on the image $I$ using a visual question answering (VQA) model, obtaining an answer alignment score $s_j = VQA(q_j, a_j, I)$ (more details on how this score is computed are given in 3.1). The overall alignment score for a image-text pair, denoted as the $VQ^2$ score, is the average over all $s_j$ scores for all the generated $(q_j, a_j)$ pairs. We next describe each step in more detail.

**Generating question-answer pairs.**    We follow the $VQ^2A$ method [17] to generate question and answer pairs given an image caption in three steps. The purpose is to generate high quality question-answer pairs, which capture the most important details of the text. First, answer spans are extracted from text $T$ using SpaCy [38], based on Part-of-Speech (POS) and dependency parse tree annotations. Then, for each answer span, a question $q_j$ is generated given the answer span and the full caption as input using a T5-XXL model fine-tuned on SQuAD1.1 [39]. Finally, each candidate question-answer pair $(q_j, a_j)$ is validated by answering $q_j$ on $T$ using a QA model, which is trained by fine tuning a

T5-XXL model on SQuAD2.0 [40] and Natural Questions [41]. Finally, we match the output answer $a'_j$ to the expected answer $a_j$ using token-level F1 comparison. As suggested in [17], if the answer comparison F1 score is lower than 0.54 the question-answer pair is filtered out.

**Assessing question-answer pair alignment against the image.** To determine if the information conveyed by the text $T$ is presented correctly in the image $I$, we use a VQA model based on PaLI-17B [42] as follows. We reformulate each question and answer candidate pair $(q_j, a_j)$ into a new yes-no predicate question $q'_j$ using the format *"is $\{a_j\}$ true for $\{q_j\}$ in this image?"*. For example, for the text *"two girls are sitting on some grass"*, and the automatically induced question-answer pair {*"what are the girls sitting on?"*, *"some grass"*}, the reformulated question is "is *on some grass* true for *what are the girls sitting on? in this image?"*. The VQA model is then invoked to answer the predicate question $(q'_j)$ over image $I$. We define the alignment score $s_j$ as the probability of the model for answering "yes". We note that we also experimented with other answer alignment methods, e.g. ones that directly ask the generated question without formulating it as a yes/no question. However, the yes-no approach worked best. More details can be found in appendix A.4.

## 3.2 End-to-end VNLI Models

Another approach is to train end-to-end Visual NLI models (VNLI) that receive an image and text as input, and directly predict an alignment score. We do so by fine-tuning multimodal pretrained models while formatting the examples as yes/no questions using the prompt: "Does this image entail the description: {text}?", followed by a binary "yes" or "no" answer. In inference time we measure the probabilities of predicting "yes" or "no", and use the relative ratio between the two as the alignment score. Specifically, we finetune BLIP2 [7] and PaLI-17B [42] using a dataset comprising 110K text-image pairs labeled with alignment annotations. This includes 44K examples from COCO-Con, 3.5K from PickaPic-Con, 20K from COCO t2i and 40K from the training split of the SNLI-VE dataset. We generate COCO-Con and COCO t2i based on the COCO train split and PickaPic-Con with a distinct set of images, to ensure that there is no overlap with samples in the SeeTRUE benchmark. More technical details and training hyperparameters are described in appendix A.7.

## 4 Experiments

### 4.1 Models and Metrics

We evaluate $VQ^2$ and fine-tuned VNLI models based on PaLI and BLIP2 (Section 3) against several state-of-the-art multimodal models: (a) CLIP [15] and two larger versions - CLIP RN50x64 and CLIP ViT-L 14 [43], (b) CoCa [22], (c) BLIP Large [6], (d) BLIP2 FlanT5-XXL [7], and (e) OFA Large [23], and (f) TIFA [44]. First five models were typically trained with either a contrastive objective or an image-text matching objective that samples positive or negative caption-label pairs. TIFA, like $VQ^2$, employs a VQA model with generated question-answer pairs. However, TIFA contrasts textual and visual answer candidates provided by the model, while our method checks if the textual answer is accurate given the image.

We assess each method's ability to detect misalignments in each dataset in SeeTRUE. We use a binary labeling scheme and report the Area Under the ROC Curve (ROC AUC) for each method. For Winoground, we use existing metrics: (1) *text score*: accuracy in selecting the right caption for an image; (2) *image score*: accuracy in choosing the correct image given a caption; (3) *group score*: accuracy requiring all four image-caption pairs to be correct for a successful example.

### 4.2 Results

We present our main results in Table 2. Notably, our $VQ^2$ approach excels as the top-performing zero-shot model across all datasets, surpassing other zero-shot baselines and even outperforming most of the fine-tuned models while achieving the highest score on the challenging Winoground dataset. This shows the robustness of the $VQ^2$ approach, which decomposes the alignment decision by generating multiple yes/no verification questions.

When looking at finetuned models, the PaLI variant finetuned on all the available datasets outperforms all the rest with an average score of 82.9, achieving the best results on 3 out of 7 datasets. The

Table 2: Main Results on SeeTRUE, split into zero-shot and end-to-end fine-tuned methods across the real and synthetic image-text eval-sets. The numbers in the table are ROC AUC. Note that TIFA and $VQ^2$ require higher computational cost. They generate question-answer pairs and use a VLM model, while the other models use the image-text directly.

| Text & Images | Real + Real | | Real + Synthetic | | | Synthetic + Real | Synthetic + Synthetic | Avg. |
|---|---|---|---|---|---|---|---|---|
| Model | SNLI-VE | Winoground | DrawBench | EditBench | COCO t2i | COCO-Con | PickaPic-Con | |
| **zero-shot** | | | | | | | | |
| CLIP RN50x64 | 66.6 | 53.6 | 59.2 | 67.1 | 58.8 | 71.1 | 66.8 | 63.3 |
| CLIP ViT-L14 | 65.8 | 53.3 | 60.5 | 62.1 | 58.8 | 70.7 | 66.8 | 62.6 |
| COCA ViT-L14 | 68.5 | 53.1 | 67.4 | 66.3 | 62.1 | 74.2 | 68.1 | 65.7 |
| COCA ViT-L14 (f.t on COCO) | 70 | 53.1 | 66.2 | 68.3 | 66.2 | 76.5 | 67.2 | 66.8 |
| BLIP | 75.2 | 58.2 | 60.5 | 68 | 70.7 | 84.2 | 76.6 | 70.5 |
| BLIP2 | 76.4 | 56.9 | 58.5 | 67.5 | 66.9 | 84.3 | 76.9 | 69.6 |
| BLIP 2 (f.t. COCO) | 75.9 | 60 | 65.7 | 70 | 73.3 | 85.8 | 78 | 72.7 |
| PaLI | 65.4 | 53.6 | 60.2 | 56.7 | 53.3 | 65.5 | 60.5 | 59.3 |
| TIFA | – | 58.0 | 73.4 | 67.8 | 72.0 | – | – | – |
| VQ$^2$ (Ours) | 88.0 | **63.5** | 82.6 | 73.6 | **83.4** | 87.1 | 81.7 | **80.0** |
| **f.t. snli-ve** | | | | | | | | |
| OFA Large (470M) | 80.5 | 53.3 | 77.6 | 70.9 | 67.5 | 75.4 | 69.5 | 70.7 |
| BLIP2 (12B) | 82.3 | 58.5 | 64.3 | 58.7 | 60.5 | 82.6 | 66.9 | 67.7 |
| PaLI (17B) | **95.1** | 61.7 | 82.8 | 65.5 | 77.7 | **91.2** | 83.7 | 79.7 |
| PaLI + Synthetic Data | 94.2 | 61.8 | **86.8** | **77.2** | 83.2 | 91 | **85.9** | 82.9 |
| Avg(VQ$^2$, PaLI+Syn) | 93.9 | **63.5** | **87.8** | **78.4** | **85.1** | **93** | **87.3** | **84.1** |

Table 3: Results on the Winoground dataset, reporting text score, image score, and group score.

| Model | Text Score | Image Score | Group Score |
|---|---|---|---|
| V$Q^2$ (Ours) | **47.00** | **42.20** | **30.50** |
| PaLI (ft SNLI-VE + Synthetic Data) | 46.5 | 38 | 28.75 |
| PaLI (ft SNLI-VE) | 45.00 | 41.50 | 28.70 |
| BLIP2 (f.t. COCO) | 44.00 | 26.00 | 23.50 |
| IAISlarge [45] | 42.50 | 19.75 | 16.00 |
| VinVL [24] | 37.75 | 17.75 | 14.50 |
| TIFA | 19.00 | 12.50 | 11.30 |
| CLIP RN50x64 | 26.50 | 13.75 | 10.25 |
| OFA Large (f.t. SNLI-VE) | 27.70 | 14.30 | 9.00 |
| COCA ViT-L14 (f.t on COCO) | 28.25 | 11.50 | 8.25 |
| Random Chance [24] | 25.00 | 25.00 | 16.67 |
| Humans [24] | 89.50 | 88.50 | 85.50 |

SNLI-VE-only variant is behind with an average score of 79.7, while achieving the highest scores for 2 out of 7 datasets. This shows that integrating synthetic training data leads to notable improvements on synthetic images on DrawBench (+4%), EditBench (+11.7%), , COCO t2i (+5.5%), PickaPic-Con (+2.2%). Nevertheless, the inclusion of synthetic training data did not enhance performance on the COCO-Con dataset, comprised solely of natural images. This indicates that the variation in image types could be a contributing factor that calls for additional exploration. Notably, the last row shows a simple average between VQ$^2$ and our leading fine-tuned PaLI model, that produces higher performance, suggesting that they complement each other effectively.

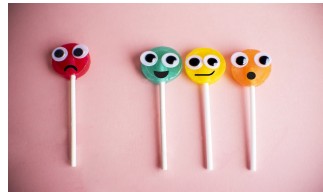
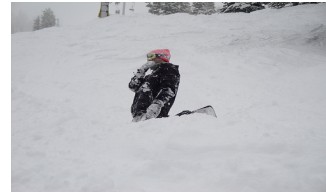
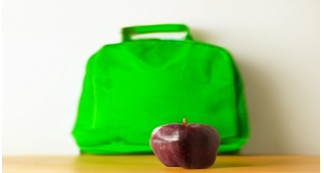

(a) "the orange lollipop is sad and the red lollipop is surprised"
Q: What is the orange lollipop feeling? A: sad
(1) Winoground

(b) "Someone in a blue hat standing on a snowy hill"
Q: What is the person wearing? A: blue hat
(2) CocoCon

(c) "A black apple and a green backpack"
Q: What color is the apple? A: black
(3) DrawBench

Figure 4: Contradicting captions and the question/answer pairs with lower $VQ^2$ alignment score, indicating the contradiction reason.

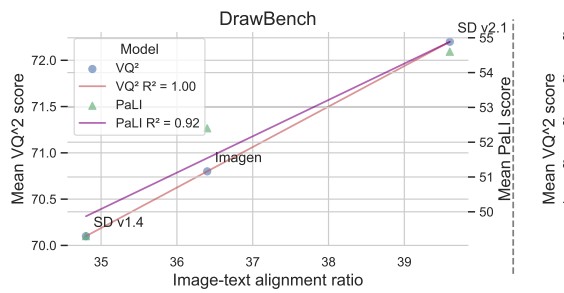 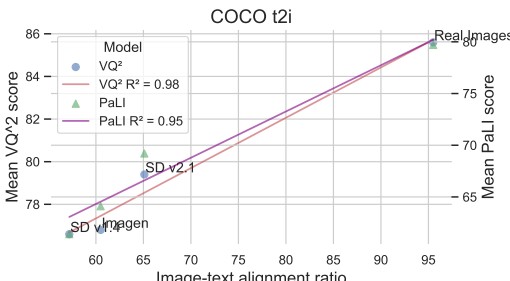

Figure 5: Highly correlated VQ$^2$ and PaLI scores vs. human rankings of text-to-image models

**Winograd Results.**    Table 3 shows the performance of the different methods on the challenging Winoground dataset, which requires strong visual reasoning and compositional understanding skills. Our zero-shot approach, VQ$^2$, achieves state-of-the-art results on this dataset, surpassing other strong baselines, with a group score of 30.5%. This again indicates that VQ$^2$'s approach that decomposes the alignment task into multiple question-answer pairs is a promising path for image-text alignment.

**Contradiction Generation.**    We assessed the VQ$^2$ method's capacity to detect image-text contradictions, as shown in fig. 4. Contradiction generation relies on identifying the question-answer pair with the lowest VQA, signaling the least likely alignment between the image and the text. Three paper authors evaluated whether a particular contradiction (consisting of a question and an answer) accurately represents the primary discrepancy between the image and the text. The majority vote among the authors determined the final outcome, yielding the following accuracy rates: 88% for Coco-Con, 74% for DrawBench, and 80% for Winoground. This indicates that our method is capable of identifying these contradictions by investigating the structure and content of the given caption and image. As a result, our method can achieve strong results, particularly on datasets that require compositional understanding.

**Comparing Generative Models.**    VQ$^2$'s ability to compare between generative models is demonstrated on the results of Draw-Bench and COCO-t2i, which include generated images from different models, together with human quality ratings. fig. 5 shows that the VQ$^2$ and our fine-tuned PaLI ranking correlates very well with human ranking ($R^2 > 0.92$). In addition, since unlike human annotations, the VQ$^2$ score is consistent across datasets, it offers a way to evaluate dataset difficulty on an absolute scale.

Table 4: Comparison of human-labeled quality scores for top-ranked images with model breakdown

| Dataset | Model | Random | CLIP | PaLI | VQ$^2$ |
|---|---|---|---|---|---|
| COCO t2i | SD 1.4 | 68.6 | 74.6 | 88.2 | 86.4 |
| | SD 2.1 | 71.3 | 81.2 | 84.5 | 87.3 |
| DrawBench | SD 1.4 | 66.7 | 77.4 | 77.4 | 87.1 |
| | SD 2.1 | 59.0 | 78.0 | 87.0 | 82.0 |

**Reranking Using Alignment Assessment.**    Alignment scores can also be used for reranking candidate generations, on top of evaluation. To demonstrate this, we re-rank the image candidate per prompt in the DrawBench and COCO-t2i datasets. We do so using VQ$^2$ and CLIP and measure the human-labeled quality of the top-ranked image for each method. The results, presented in table 4, show that ranking with VQ$^2$ consistently achieves higher quality scores when compared to ranking with CLIP. One such example is shown in fig. 6, where both VQ$^2$ and our top-performing fine-tuned PaLI model demonstrate superior ranking by placing the brown-and-white cats above the white-only cats. This consistency between VQ$^2$ and PaLI highlights their alignment evaluation models' potential for enhancing text-to-image systems, which contrasts with the divergent ranking exhibited by CLIP.

**A brown and white cat is in a suitcase**

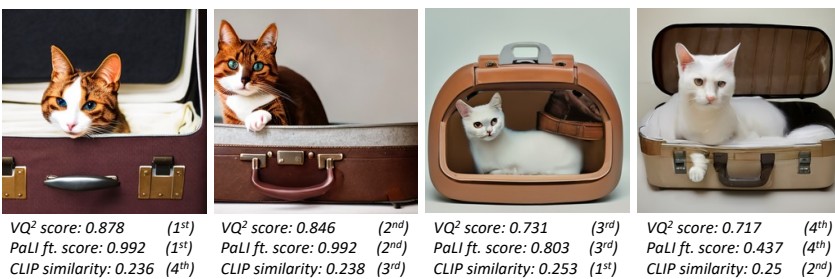

| VQ² score: 0.878 | (1st) | VQ² score: 0.846 | (2nd) | VQ² score: 0.731 | (3rd) | VQ² score: 0.717 | (4th) |
| PaLI ft. score: 0.992 | (1st) | PaLI ft. score: 0.992 | (2nd) | PaLI ft. score: 0.803 | (3rd) | PaLI ft. score: 0.437 | (4th) |
| CLIP similarity: 0.236 | (4th) | CLIP similarity: 0.238 | (3rd) | CLIP similarity: 0.253 | (1st) | CLIP similarity: 0.25 | (2nd) |

Figure 6: Four COCO-t2i text-to-image model outputs ranked by VQSQR scores, correlating with top PaLI model. Image order and CLIP (RN50x64) similarity scores given, but not aligned with VQ²/PaLI ranks.

## 5    Related Work

Our work advances research in visual entailment (VE) [27], visual question answering (VQA) [46], text-to-image alignment evaluation, and cross-task consistency for multi-modal models, with a focus on enhancing the semantic understanding of image-caption relationships.

Textual Entailment (TE) [26, 47] evaluates the truthfulness of a textual hypothesis given a textual premise, providing a key benchmark for the semantic capabilities of neural network models [48–50, 35]. Recently, TE has been adapted to the multimodal domain as Visual Entailment (VE) [27] to assess the semantic alignment between images and text. Vision-and-language models like CLIP [15], CoCa [22], BLIP [6], BLIP2 [7] and OFA [23] often act as bag-of-words models, lacking a deep comprehension of language compositionality [16]. Our approach addresses this by generating multiple questions probing diverse semantic aspects, thereby improving performance on challenging compositional tasks like Winoground [24] and unnatural images as in Drawbench [5].

Unlike DrawBench [5] and DALL-Eval [51] which depend on human feedback and operate within a discrete set of alignments, our approach produces automated scores for a broader range of text-image alignments, facilitating efficient evaluation of vision-and-language models. Our approach also surpasses the recently proposed TIFA [44], which may be due to employing more question-answer pairs and tailored models for question generation and answering.

Several works have explored cross-task consistency in multi-modal models across various modalities. VQA studies have tackled inconsistencies and enhanced consistency using data augmentation and contrastive loss. NLP researchers have improved consistency across tasks or within a single task by employing counterfactual instances or contrast sets [26, 47]. Our research aligns with studies that evaluate natural text and images [52]; however, extending the focus to synthetic images and texts, and aligning with synthetic image understanding research [53–58]. We introduce two unique approaches to address the complexities of image-text alignment.

Another related effort is PickScore [29], which predicts human preferences for image quality and aesthetics by ranking or choosing between two images. In contrast, our methods independently score a single image and focus specifically on image-text alignment.

## 6    Limitations

We recognize that in some cases, making a binary decision for whether a text and an image are aligned may be be difficult, also for humans. To tackle this limitation, we provided human annotators with comprehensive guidelines, which resulted in a high inter-annotator agreement (Fleiss-Kappa score of 0.722 with 80% of the cases where all annotators agreed on the entailment label).

Although many images in our datasets were obtained by others and not created by us, we made an effort to ensure that they do not contain harmful or potentially harmful content, such as NSFW or biased imagery. During the annotation process, three individuals examined each image and indicated if it could be considered offensive. Additionally, two of the authors manually reviewed the images

for any harmful content. However, we understand that the perception of harmful or offensive content may vary among individuals and may be subject to personal interpretation.

## 7 Conclusion

We addressed the task of image-text alignment evaluation, which we find very close to the Visual Entailment (VNLI) task. We first introduced the SeeTRUE benchmark, which covers the mix of real and synthetic text and image pairs in text-to-image and image-to-text generation tasks, and includes challenging cases based on generated contradictions. We then proposed two methods, $VQ^2$ and end-to-end VNLI, which outperform strong baselines on SeeTRUE and can serve as a starting point for future research on the task.

In future work, we would like to employ our automatic evaluation models for guiding the training of text-to-image and image-to-text models towards more aligned outputs, following recent trends in text-to-text generation [59, 60]. For example, such models may be useful either for filtering training examples or as a reward when training models using reinforcement learning.

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

# A  Appendix

## A.1  Dataset Supplementary Materials

1. Dataset documentation, metadata, and download instructions: `anonymous`.

2. Intended uses: we hope SeeTRUE will be used by researchers to evaluate image-text matching models.

3. Author statement: We bear all responsibility in case of violation of right in using our benchmark.

4. Each dataset's license is described below. Our additional human annotations and geen-rated images are licensed under CC-BY 4.0 license `https://creativecommons.org/licenses/by/4.0/legalcode`.

5. Hosting & preservation: the dataset will be hosted in Huggingface Datasets, accessible and available for open research.

6. SeeTRUE fields are presented in table 5.

Additional licensing details: we do publish the datasets we annotated in this work, and do not re-publish the existing SNLI-VE and Winoground datasets. Full licenses:

1. MS COCO [28]: `https://cocodataset.org/#termsofuse`

2. EditBench [25]: `https://research.google/resources/datasets/editbench/`, `https://www.apache.org/licenses/LICENSE-2.0`

3. DrawBench [5]: `https://imagen.research.google/`, `https://docs.google.com/spreadsheets/d/1y7nAbmR4FREi6npB1u-Bo3GFdwdOPYJc617rBOxIRHY/edit#gid=0`

4. Pick-a-Pick [29]: `https://huggingface.co/datasets/yuvalkirstain/pickapic_v1`

5. SNLI-VE [27]: `https://github.com/necla-ml/SNLI-VE`

6. Winoground [24]: `https://huggingface.co/datasets/facebook/winoground`

Table 5: SeeTRUE Rows Examples

| image | text | label | original_dataset_id | dataset_source |
|-------|------|-------|---------------------|----------------|
| img1 | A zebra to the right of a fire hydrant. | 0 | text_133_image_1228 | drawbench |
| img2 | A group of people standing next to bags of luggage. | 1 | text_105_image_1377 | coco_t2i |
| img3 | a tiny figurine is surrounded by cell phones on a table. | 1 | 3786 | editbench |

## A.2  Additional Examples

We added additional examples from (a) instances from SeeTRUE fig. 7; (b) the $VQ^2$ method fig. 8 to visualize the question-answer pairs; and (c) the ConGen technique fig. 9. This will underscore the versatility of ConGen in handling both real and synthetic images, as well as the variety of contradictions it can generate.

## A.3  Human Annotation Process

In order to provide reliable human labels for our datasets, we conducted an annotation process using the SeeTRUE platform. The process comprised several steps, including setting qualification requirements, providing instructions, and evaluating annotator agreement.

We set the basic requirements for our annotation task as follows: a percentage of approved assignments above 98%, more than 5,000 approved HITs, and annotator locations limited to the US, UK, Australia, or New Zealand. We selected 5 examples from our dataset for a qualification test and screened the annotators' results. fig. 10 displays a sample of the Mechanical Turk user interface. The payment for the crowd-workers was 15-18 USD hourly.

| image (image) | text (string) | label (int64) | original_dataset_id (string) | dataset_source (string) |
|---|---|---|---|---|
| | "A zebra to the right of a fire hydrant." | 0 | "text_133_image_1228" | "drawbench" |
| | "A group of people standing next to bags of luggage." | 1 | "text_105_image_1377" | "coco_t2i" |
| | "The old tow truck tries to lift an overturned truck…" | 0 | "text_178_image_2319" | "coco_t2i" |
| | "Abraham Lincoln touches his toes while George Washington…" | 0 | "text_103_image_993" | "drawbench" |
| | "A cat on the lid of a toilet looking perturbed." | 1 | "text_3_image_47" | "coco_t2i" |

Figure 7: An example from SeeTRUE Huggingface Dataset.

**A brown bird and a blue bear**

| Question | Answer (based on text) | VQA "yes" probability | VQ2 score |
|---|---|---|---|
| Aside from a blue bear, what other animal is depicted in the picture? | A brown bird | 0.26 | |
| Aside from the brown bird, what other animal is in the picture? | a blue bear | 0.41 | |
| What is the blue animal? | bear | 0.44 | 0.4 |
| Aside from a bear, what other animal is depicted? | bird | 0.37 | |
| What color is the bear? | blue | 0.5 | |
| What color is the bird in the picture? | brown | 0.42 | |

**A wine glass on top of a dog**

| Question | Answer (based on text) | VQA "yes" probability | VQ2 score |
|---|---|---|---|
| What is on top of a dog? | A wine glass | 0.82 | |
| A wine glass is on top of what animal? | a dog | 0.78 | |
| What is the wine glass on top of? | dog | 0.77 | |
| What is on top of a dog? | glass | 0.74 | |
| On what part of the dog is the wine glass located? | top | 0.64 | 0.78 |
| Where is the wine glass located? | top of a dog | 0.88 | |
| What kind of glass is on the dog? | wine | 0.82 | |
| What is on top of a dog? | wine glass on top | 0.79 | |

Figure 8: Examples of the $VQ^2$ method. The answers extracted from the text are named entities, nouns, and multi-word spans, such as adjectives and locations.

The instructions provided were as follows:

*Evaluate the given image and text to determine if they match, selecting either "Yes" or "No". Some images may be synthetically generated by a text-to-image model. To assess the match, mentally generate a textual description for the image (no need to write it down) and compare this generated description to the given text. If the descriptions closely resemble each other, mark "Yes". If not, mark "No" and provide feedback on the specific issue causing the misalignment, focusing on the primary issue if multiple misalignments are present. If you encounter an image or text that may be offensive due to bias, race, NSFW content, etc., mark the checkbox to indicate this issue.*

Full agreement metrics are presented in table 6. As shown in the table, the percentage of cases where all annotators agreed and the Fleiss-Kappa scores vary across the datasets, with COCO-Con

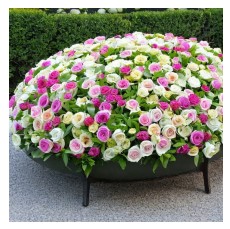
large white and pink rose planter surrounded by **lush** greenery

large white and pink rose planter surrounded by **dead** greenery

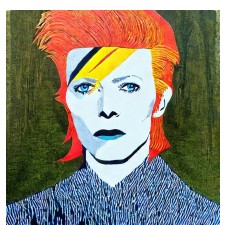
a **red** haired bowie is painted on the wall

a **blue** haired bowie is painted on the wall

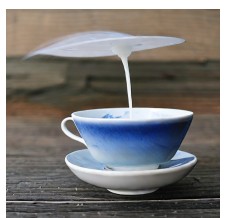
milk pouring into a coffee cup on a **wooden** table

milk pouring into a coffee cup on a **plastic** table

Figure 9: Examples from the ConGen method to generate contradictions. The top caption is the original, positive one, and the bottom one is the contradiction generated by ConGen.

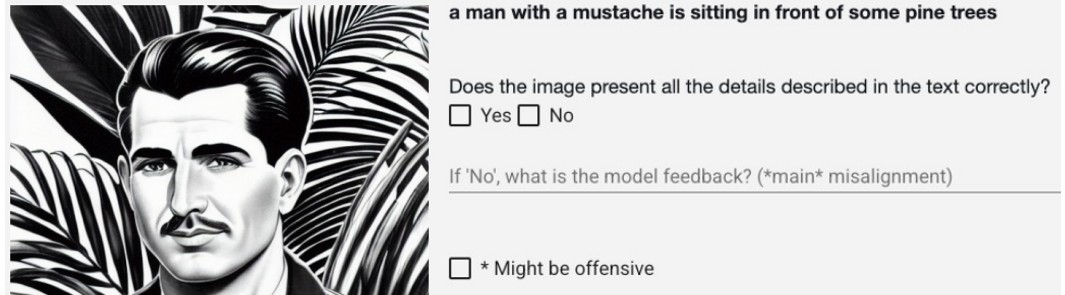

Figure 10: Annotation interface for determining whether a given image-text pair are aligned.

exhibiting the highest level of agreement and Drawbench the lowest. These differences highlight the varying levels of complexity within the datasets.

Table 6: Agreement metrics for different datasets.

| Dataset | Full | Drawbench | COCO t2i | COCO-Con | PickaPic-Con |
|---|---|---|---|---|---|
| # Items | 8,527 | 1,968 | 2,586 | 1,992 | 1,981 |
| % all agreed | 80 | 76 | 78 | 86 | 77 |
| Fless-Kappa | 0.72 | 0.66 | 0.68 | 0.81 | 0.69 |

## A.4 Comparing $VQ^2$ variants

$VQ^2$ consists of two main parts: generating question-answer pairs and assessing question-answer pair alignment against the image. We have experimented with different variants of the $VQ^2$ zero-shot method.

**Assessing question-answer pair alignment methods**   Given a question-answer pair, we would like to assess the question based on the image and compare it to the information in the text. We experimented with several configurations for answer alignment:

1. Type A: Given a question-answer pair $(q_j, a_j)$ generated from the text, we answer the question using a VQA model and obtain an answer based on the image $a_j^I = VQA(q_j, I)$. We compare a pair of $(q_j, a_j)$ with a pair of $(q_j, a_j^I)$ using an Natural Language Inference (NLI) model, where the pair based on the text serves as the premise and the other as the hypothesis. We define the alignment score $s_j$ as the probability of the NLI model for answering "entailed".

2. Type B: As done in type A, we answer the question using a VQA model and obtain the answer $a_j^I = VQA(q_j, I)$. We use the VQA model again to compare the the two answers and determine whether they are the same. The question is formulated as *"Is $\{a_j\} == \{a_j^I\}$ in this image?"*. We define the alignment score $s_j$ as the probability of the VQA model for answering "yes".

3. Type C: We reformulate each question and answer candidate pair $(q_j, a_j)$ into a new yes-no predicate question $q_j'$ using the format *"is $\{a_j\}$ true for $\{q_j\}$ in this image?"*. The VQA model is then invoked to answer the predicate question $(q_j')$ over image $I$. We define the alignment score $s_j$ as the probability of the model for answering "yes".

**Generating question-answer pairs**   To produce question-answer pairs from the text, we first extract informative spans in the text $T$. We extract as answer candidates all named entities and noun phrases in $T$ using spaCy. We noticed that for short text $T$, this method doesn't produce enough question-answer pairs to assess the alignment between the text and the image. Thus, we extend the answer candidates by adding multi-word spans, such as adjectives ("black and white") and location "in the air". We use the extended answer candidate extraction in all of our experiments. The number of question-answer pairs is dependent on the length of the text T. In EditBench, the text contains 12.5 words on average, and it results in average of 34.9 question-answer pairs with extended answer candidates.

Table 7 summarize the results of the $VQ^2$ variants on the EditBench dataset. Our zero-shot $VQ^2$ method is $VQ^2$ type C, it outperforms other configurations and it is more efficient, since it requires a single run of the VQA model for the question-answer pair assessment.

Table 7: Comparing $VQ^2$ configurations on all EditBench categories

| Method | Models used for assessment | **EditBench** | | | | | |
|---|---|---|---|---|---|---|---|
| | | Color | Count | Material | Shape | Size | Mean |
| $VQ^2$ (A) | VQA & NLI | 75.7 | 63.6 | 71.9 | 67.6 | 75.9 | 70.9 |
| $VQ^2$ (B) | VQA & VQA | **80.2** | 72.3 | 75.6 | **73.5** | 75.4 | 75.4 |
| $VQ^2$ (C) w.o. multi-word answers | VQA | 77.2 | 71.5 | 73.6 | 71.9 | 75.5 | 73.9 |
| $VQ^2$ (C) | VQA | 78.5 | **73.5** | **76.9** | 71.7 | **78.2** | **75.8** |

## A.5   Evaluating Contradictions Generated by GPT4

To assess GPT4's capability in generating contradictions, we compared the outputs from a GPT4-driven ConGen approach against our PaLM-based method. We selected a set of 100 images from both COCO-Con and PickaPic-Con databases. For each image, we obtained 100 positive and 100 negative captions using PaLM. Subsequently, by utilizing the identical prompt with GPT4, we generated an additional 100 negative captions. These captions were randomly assigned labels 'A' and 'B'.

For the evaluation process, we engaged three independent evaluators from Amazon Mechanical Turk. They were presented with an image, its caption, and the two negative caption candidates. Their task was to discern whether each candidate accurately contradicted the depicted image. A snapshot of this user interface can be seen in fig. 11.

The final judgment was based on the majority vote of the workers. The results showed a close match in performance between the two models: PaLM achieved 77% while GPT4 reached 76%. This suggests our ConGen approach works effectively with various large-scale language models.

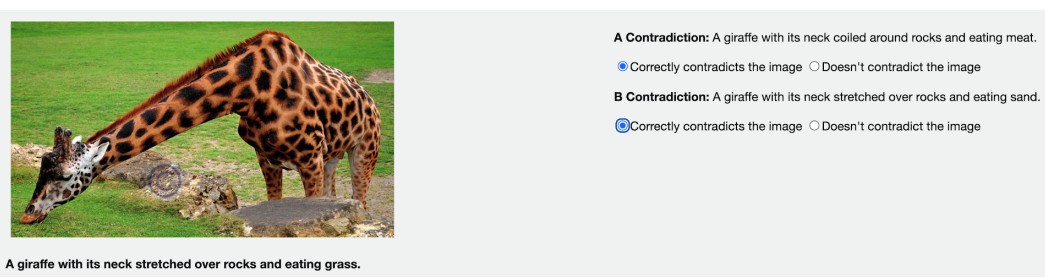

Figure 11: The Amazon Mechanical Turk user interface on determining whether a contradicting caption generated by GPT4 or PaLM is correct.

## A.6 Comparing $VQ^2$ with PaLI vs. BLIP2

We compare PaLI and BLIP2 for the VQ$^2$ method in table 8. While PaLI yields higher scores, it's pivotal to recognize their distinct evaluation metrics. PaLI's score indicates the probability of a "yes" response, whereas BLIP2 gives a definitive answer without probability. This difference can influence the application suitability of each model.

|  | PaLI | BLIP2 |
|---|---|---|
| Drawbench | 82.6 | 60.5 |
| Winoground | 63.5 | 58.2 |
| COCOCON | 87.1 | 88.5 |

Table 8: Comparison of VQ$^2$ results for BLIP2 and PaLI (ROC AUC scores). Note the difference in evaluation: PaLI's VQ$^2$ score represents the probability of a "yes" response, while BLIP2 offers a definitive answer without probability.

## A.7 Reproducibility

To fine-tune BLIP2, we adjust only the Q-former parameters of the model using the Adam optimizer. We train the model for two epochs and designate 10% of the training set as a validation set for early stopping and use learning rate selection between {1e-5, 5e-5}. A single training took 5 hours on a linux server with one A6000 GPU. All experiments took <2 days.

Zero-shot $VQ^2$: For 10,000 text-image pairs, the inference time of every step is as follows. Answer candidate generation: when using extended answer candidates – about 1 day. Otherwise, 12 hours. Question generation and filtering: When using extended answer candidates, about 2 days, otherwise, 1 day. The last step only takes a few minutes.

| Aspect | PaLI | $VQ^2$ | BLIP2 |
|---|---|---|---|
| Inference Time | 500ms per image-text pair | 40 seconds per image (full pipeline) | 750ms per image-text pair (as measured in source) |
| Model Parameters | 17B parameters | T5-XXL - 11B parameters + PaLI 17B | 12B parameters |
| Hardware Requirements | Four v4 chips [61] | 16 TPU v3 cores + 4 v4 chips [61] | GPU with 24GB as reported in HuggingFace |
| Framework | T5X [62] on JAX [63] | T5X [62] on JAX [63] | Pytorch |

Table 9: Computational Costs Summary

