# OpenReview forum: "What You See is What You Read? Improving Text-Image Alignment Evaluation"
_NeurIPS.cc/2023/Conference — NeurIPS 2023 poster_

### Official Review · Reviewer_UX4d · 2023-07-04

**Soundness:** 3 good
**Presentation:** 3 good
**Contribution:** 3 good
**Rating:** 6
**Confidence:** 5

**Summary:**

This paper aims to develop a method for evaluating the level of semantic alignment between text and image. In order to achieve this, the authors construct datasets of image and text pairs, and collect human judgments to determine if the text and image are semantically aligned. The paper proposes two methods for estimating alignment between text and image. The first method generates questions and answers from text and checks whether a Visual Question Answering (VQA) model provides consistent answers to the questions. The second method is an end-to-end classifier that predicts if the image entails the text. Experimental results demonstrate that the proposed methods outperform existing multimodal models.

**Strengths:**

- A significant contribution of this paper is the collection of human judgments on text and image alignment. This will aid in the development and validation of alignment evaluation methods in future research.
- The annotation pipeline is carefully designed, and the details are throughly described.

**Weaknesses:**

- The VQ^2 method depends on the sampling of questions and answers, which may result in output variation due to chance. The effects of sampling are not discussed in the paper.
- Language models used to generate contradicting captions may introduce biases in the datasets. For example, generated captions might have typical wording patterns that are difficult for humans to detect. Validating synthesized datasets is challenging.

**Questions:**

- Can the authors provide more details about the PaLM instruction used to generate contradicting captions? Was the prompt in Figure 2 utilized to rewrite captions?

**Limitations:**

The paper discusses two limitations:
1. The difficulty in judging alignment between a text and an image even for humans.
2. The challenge of filtering offensive content from the dataset.

---

> ### Author Rebuttal · Authors · 2023-08-08
>
> We thank the reviewer for acknowledging the value of our human image-text judgments, that will aid in the development and validation of alignment evaluation methods.
>
> ### Variability in $VQ^2$ Sampling [weakness 1]
>
> Our strategy for question-answer pair generation is based on established works like $Q^2$ and $VQ^2A$. We choose answer candidates by extracting noun phrases using the SpaCy toolkit so they are not random. For the QG we do beam-search, not sampling, but generating 5 questions per answer and filter out low quality question and answer pairs.
>
> ### Validating contradictions generated by the LLM [weakness 2]
>
> For our SeeTRUE test set, the labels are a product of a majority consensus among three annotators. This yielded a high inter-annotator agreement, as documented in Table 6. Such consistency among annotators signifies a shared understanding of the semantics inherent in the generated captions. We agree that generating the contradictions with different LLMs would reduce biases of a single LLM. In light of your and reviewer j32y feedback, we tested GPT4 for Generating Contradicting Captions, which yielded comparable results to our initial model. We believe that the combination of several LLMs and VLMs in the ConGen methodology reduces biases introduced by a single LLM.
>
> When considering the training set, we acknowledge that it contains a proportion of auto-labels. However, as depicted in Table 6, any noise introduced is minimal. More crucially, even when such noise is present, Table 2 shows that fine-tuning on this synthetic dataset leads to enhanced performance on SeeTRUE, which employs human-annotated labels.
>
> ### PaLM instruction [question 1]
> You're right. The prompt illustrated in Figure 2 is precisely what we deployed to produce contradictions. This was done by running the prompt with multiple few-shot examples, like the three presented in Figure 2. By altering specific elements (e.g., "knife" to "spoon" or "rainforest" to "desert"), we applied this to the five input captions using the LLM. This procedure is adaptable and can integrate other LLMs like GPT4 or Stable Beluga, among others. Once the contradictions were generated, we employed an NLI model to rank each one, selecting the “most contradicting” candidate which is identified by the lowest NLI score.

---

> > ### Comment · Reviewer_UX4d · 2023-08-12
> > **Thanks for your response**
> >
> > Thank you for your responses.
> > The rebuttal provides clarification to my questions.
> > I am increasing my score.

---

### Official Review · Reviewer_xP6K · 2023-07-05

**Soundness:** 3 good
**Presentation:** 3 good
**Contribution:** 3 good
**Rating:** 7
**Confidence:** 3

**Summary:**

This paper addresses the problem of text-image alignment evaluation by proposing a benchmark named SeeTRUE and two alignment metrics named VQ^2 and end-to-end VNLI.

SeeTRUE covers a variety of real and synthetic images and captions. The synthetic captions, which are generated by large language models (LLMs), preserve most words from the original captions but convey a contradictory meaning. The synthetic images are generated by text-to-image generative models. The real image and captions are collected from existing datasets. In addition to the image-text pairs, SeeTRUE also provides binary human annotation of the alignment between these pairs.

VQ^2 first generates question-answer pairs based on the caption and then assesses question-answer pair alignment against the image. End-to-end VNLI directly fine-tunes a visual NLI model to determine whether an image entails a caption.

Experiments show that the proposed image-text alignment metrics outperform state-of-the-art (SOTA) vision-language models on the SeeTRUE benchmark. The VQ^2 metric also exhibits (1) a high correlation with humans in the evaluation of text-to-image generative models and (2) strong potential as a reranking technique to select generated images.


**Strengths:**

* The proposed SeeTRUE overcomes two limitations found in existing benchmarks: a primary emphasis on real images and a lack of challenging negative captions. SeeTRUE provides a valuable testbed for evaluating image-text alignment methods.
* The proposed two image-text alignment metrics are well-designed and exhibit very promising results, which can facilitate the evaluation and development of image captioning and text-to-image generation models.
* The experiments are comprehensive and well-designed.
* The paper is well-written and easy to follow.


**Weaknesses:**

The current manuscript still has three problems (which do not warrant rejection):
* SeeTRUE primarily focuses on the challenging scenario where negative captions differ from positive captions by only a few words. However, it remains unclear whether the proposed VQ^2 and End-to-end VNLI models, specifically tailored for this setting, can still surpass the strong baselines in the standard scenario where negative captions describe completely different images.
* Considering the utilization of three models—namely, an answer generation model (T5-XXL), a QA model (T5-XXL) and a VQA model (PaLI-17B)—VQ^2 might incur higher computational costs compared to the baseline methods. It would be beneficial to provide a detailed account of the computational expenses of these methods in Table 2.
* It seems that the results in Table 7 of Appendix A3 are missing.


**Questions:**

N/A

**Limitations:**

Please refer to the “Weakness” section.

---

> ### Author Rebuttal · Authors · 2023-08-08
>
> We thank the reviewer for acknowledging that our methods can facilitate the evaluation and development of image captioning and text-to-image generation models.
>
> ### Evaluation Scenarios with Different Negative Captions [weakness 1]
>
> SeeTRUE's diverse architecture encompasses seven distinct test sets. Three of them, namely Winoground, COCO-Con, and PickaPic-Con, operate on a "few-words" change dynamic. Conversely, the remaining datasets, such as SNLI-VE, DrawBench, EditBench, and COCO t2i, explore alternative evaluation angles. For instance, SNLI-VE extracts text descriptions from *entirely distinct images*, while COCO t2i offers a *broader semantic spectrum* due to its text-image generation capability. These variations ensure SeeTRUE’s capacity to evaluate alignment over a wide range of text perturbations, and not just simple lexical changes.
>
> ### Computational Costs of $VQ^2$ vs. VNLI [weakness 2]
>
> Indeed, $VQ^2$’s sequential pipeline can present more computational overhead compared to simpler models like VNLI. However, this modular approach is advantageous for a few reasons:
> 1. **Robustness**: As a zero-shot model, $VQ^2$ steers clear of shallow heuristics during training, ensuring a comprehensive evaluation.
> 2. **Interpretability**: The $VQ^2$ model can effectively pinpoint the questions that majorly contribute to misalignments. This is evident in Figure 4.
> 3. **Data Generation Capability**: $VQ^2$ doubles as a synthetic data generation tool. Typically employed offline, it can generate pertinent questions and answers for datasets consisting of images and texts. As we work on optimizing its latency, we're confident of further enhancing its efficiency.
>
> We are continuously working to refine the computational efficiency of $VQ^2$, ensuring its alignment with real-world applications.
>
> ### Table 7 appendix missing results [weakness 3]
>
> We appreciate you pointing this out. We have since addressed and rectified this oversight.

---

> > ### Comment · Reviewer_xP6K · 2023-08-17
> > **Response to Author Rebuttal**
> >
> > Thanks for your response. My first concern is well addressed. As for the second concern, I agree that sequential pipeline is advantageous in the three aspects. However, I think it would be better to report the computational cost (e.g., time cost and the amount of GPU memory used) in the main results of Table 2.

---

> > > ### Author Response · Authors · 2023-08-17
> > > **Computational Costs**
> > >
> > > Thank you for your feedback. In response to your suggestion, we have now incorporated a detailed computational cost section within the paper. Below is a brief summary:
> > >
> > > | Aspect               | PaLI                                | $VQ^2$                                        | BLIP2                                                                                                               |
> > > |----------------------|-------------------------------------|------------------------------------------------|---------------------------------------------------------------------------------------------------------------------|
> > > | Inference Time       | 500ms per image-text pair           | 40 seconds per image (full pipeline)           | 750ms per image-text pair (as measured in [source](https://arxiv.org/abs/2303.11897))                                 |
> > > | Model Parameters     | 17B parameters                      | T5-XXL - 11B parameters + PaLI 17B             | 12B parameters                                                                                                      |
> > > | Hardware Requirements| Four v4 chips (Jouppi et al., 2020) | T5-XXL: 16 TPU v3 cores + PaLI: 4 v4 chips                   | GPU with 24GB as reported in [HuggingFace](https://huggingface.co/spaces/Salesforce/BLIP2/discussions/2#:~:text=The%20hardware%20requirements%20depend%20on,up%20to%2024Gb%20during%20inference.) |
> > > | Framework            | T5X (Roberts et al., 2022) on JAX (Bradbury et al., 2018) | T5X (Roberts et al., 2022) on JAX (Bradbury et al., 2018) | Pytorch                                                                                                                     |

---

### Official Review · Reviewer_prMg · 2023-07-07

**Soundness:** 3 good
**Presentation:** 3 good
**Contribution:** 3 good
**Rating:** 6
**Confidence:** 4

**Summary:**

The authors propose a benchmark and two methods for evaluating fine-grained and complex image-text alignment. Their benchmark involves multiple distracter captions and images involving both real and synthetic images and captions. They propose two methods - one evaluates image-text alignment by asking multiple entailed questions, and another is a large Vision-Language model fine-tuned on their benchmark image-caption data.

**Strengths:**

Evaluating by visual entailment is neat since it is interpretable.
They also show that they outperform a contemporary benchmark TIFA that does the same.
They release a comprehensive benchmark, a combination of available data and annotations they collect, which will be valuable to the community for evaluating vision-language models.

**Weaknesses:**

Sometimes certain design choices are not well motivated and make the reader wonder why such a choice was adopted. For example, when generating qa pairs, the question is answered using a t5 Language model first to filter some qa pairs. Is this to ensure the answer semantically makes sense for such a question in general? If so, writing out clear motivation before explaining what is being done might be good.

The benchmark seems to be a collection of simply more human-annotated image-text data. Motivating why sometimes synthetic images or text is required and what are new ways one can use this to evaluate their models (that current benchmarks lack) would be nice. For instance, with the contrastive captions, one can evaluate compositionality. However, we can already do this with Winoground; what extra does this benchmark give us?

Some other relevant image-text benchmarks that evaluate compositionality using similar ideas of distractor captions are CREPE (https://arxiv.org/abs/2212.07796) and COLA (https://arxiv.org/abs/2305.03689). It might be good to add in a discussion of how this benchmark relates.

(minor) Even though synthetic images and texts are human evaluated and filtered, the datasets may be biased to having images that current generation models can already generate well, limiting the application of evaluating image generation models on the dataset on edge case prompts.

**Questions:**

What are the train, val, and test splits of SeeTRUE? If I understand correctly, the VQ2 models are not trained and are just evaluated on the test split. However, the VNLI is trained on the train splits of SeeTRUE and then evaluated on the test splits. Is that correct?

How many questions are asked for a given image-text pair in the VQ2 method?

**Limitations:**

Authors discuss limitations sufficiently

---

> ### Author Rebuttal · Authors · 2023-08-08
>
> We thank the reviewer for recognizing our approaches and the SeeTRUE benchmark as valuable to the community for evaluating vision-language models. We will address your comments and questions.
>
> ### Clarity on $VQ^2$ Design Choices [weakness 1 and question 2]
>
> We've elaborated on the motivations underlying $VQ^2$ in Section 3.1. Further, Appendix A.3 offers insights into other $VQ^2$ variants we explored.
>
> Our strategy for question-answer pair generation is based on established works like $Q^2$ and $VQ^2A$. We leverage a Question Generation (QG) model, and to refine the output, a Question Answering (QA) model is deployed against the text, thereby filtering irrelevant or low-quality questions.
>
> ### Benchmark's Value as a Comprehensive Dataset [weakness 2&3 and question 1]
>
> Our benchmark's unique blend of natural & synthetic images and text differentiates it from Winoground, CREPE, and COLA. This composition permits more robust system evaluations. While Winoground adeptly assesses compositionality, our approach, with its contrasting captions, hones in on finer details such as color, objects, and image composition.
>
> Notably, advanced chatbots like Bing Chat now serve users by either sourcing existing images or generating new ones “on the fly”. This evolution underscores the importance of real text and prompts in contemporary evaluations. Furthermore, there's a shift towards synthetic/generated captions in academia, highlighted by studies like "Improving Multimodal Datasets with Image Captioning" (https://arxiv.org/abs/2307.10350). Such research showcases how machine-generated captions can even outperform human annotations, thus elevating performance across diverse tasks.
>
> Our SeeTRUE benchmark, particularly with synthetic datasets like DrawBench and EditBench, presents genuine challenges to image-text alignment methodologies. By using various text-to-image models, several data sources, and diverse prompts, we aim for a rich image-text example distribution. The inclusion of human-annotated pairs ensures our benchmark challenges text-to-image models with nuanced misalignments.
>
> ### Details on Training & Testing [question 1]
>
> Our primary objective for SeeTRUE was to establish a high quality, diverse benchmark for image-text alignment. As you've rightly noted, while the $VQ^2$ method undergoes evaluation on the test split, we utilize SNLI-VE train and validation splits for the VNLI model. Our training set further benefits from additional data, detailed in section 3.2 and Appendix A.4.

---

### Official Review · Reviewer_j32y · 2023-07-25

**Soundness:** 2 fair
**Presentation:** 2 fair
**Contribution:** 2 fair
**Rating:** 6
**Confidence:** 3

**Summary:**

This paper introduces SeeTRUE, a benchmark for evaluating image-text alignment, encompassing a diverse range of both real and synthetic images and text. The authors proposes two innovative approaches to evaluating alignment: VQ2, which relies on question generation and visual question resolution, and VNLI, which relies on fine-tuning substantial multimodal language models. The suggested methods perform better on a variety of alignment tests than earlier methods, particularly in difficult scenarios involving complicated composition or strange images. The study also shows how these techniques may rerank generated image candidates and pinpoint particular misalignments. This provides alignment evaluation methods for image-to-text and text-to-image models.

**Strengths:**

- This paper introduces comprehensive benchmark, SeeTRUE covers real and synthetic text and image pairs from a variety of different tasks, allowing for a more thorough evaluation of text-image alignment models. This could be a solid contribution on this field.
- The paper presents two new text-image alignment evaluation techniques, VQ2 and VNLI. these methods outperform previous ones, especially in complex or unnatural images.
- The paper is clearly written, and easy to follow
- The author shared part of their codes

**Weaknesses:**

- Because performance is dependent on the LLM, the comparison doesn't seem fair.
- Also, as Generating Contradicting Captions (even with LLM) isn't a new concept, there isn't much insight to be gained from this paper.
( for example, Momeni et al. https://arxiv.org/abs/2304.06708  )
- There aren't many examples. Even the appendix is not sufficient. It would be better if there were more examples.


**Questions:**

-Regarding the Dependency on the LLM, the methods largely depends on the quality of the LLM and language-and-vision model. Does this mean that the proposed methods would perform poorly if the model is not good enough? Can you provide a comparison of how these methods perform with different LLM and VLM models?

-On the Novelty of Generating Contradicting Captions, Since the generation of contradicting captions is not a new concept, how does the proposed negative mining with LLM methods in this field? Is there an additional aspect of the method that brings unique contributions to the field of text-image alignment?

-Can these methods be adapted for other tasks that involve the interaction of text and image

**Limitations:**

The author addressed the limitations of this work. The authors could discuss the limitation of prompting LLM in more detail, as well as suggest ways to mitigate it.

---

> ### Author Rebuttal · Authors · 2023-08-08
>
> We thank the reviewer for acknowledging the contribution of the SeeTRUE benchmark to allow a more thorough evaluation of text-image alignment methods, and for recognizing our approaches as innovative. We will now address your comments and questions.
>
> ### Dependency on LLM/VLM Quality + LLM Limitations [weakness 1 and question 1 + limitation]
>
> Our ConGen method does depend on an LLM, but with the increasing availability of robust LLMs like GPT4, Stable Beluga, Vicuna, Falcon, LLAMA2 etc. our methodology can be applied by others.
>
> Based on your feedback, we explored the capability of GPT-4 in producing Contradicting Captions. We conducted an evaluation on Amazon Mechanical Turk, engaging three annotators to ascertain if the contradiction candidates correctly contradicts the image. The results are on par with our original model, indicating that our ConGen method is compatible and effective with other leading language models. The full analysis is presented in Appendix A.5 in the revised paper.
>
> As per VLM models, our results (Table 2) illustrate that the VNLI model based on PaLI excels over the BLIP-2-based model, where both are trained on the SNLI-VE dataset. Additionally, the quality of VNLI can be further improved with more extensive fine-tuning.
>
> We also added the results of using VQ$^2$ with BLIP-2 instead of PaLI in Appendix A.6. Regarding the QG and QA models used in the V$Q^2$ method, in the Q$^2$ work (https://arxiv.org/abs/2104.08202) Table 8 presents results with T5-small instead of T5-XXL (11B) and the results are inferior.
>
> ### Generating Contradiction Captions [weakness 2 and question 2]
>
> While prior works like BLIP2 (https://arxiv.org/abs/2301.12597) and ALIGN (https://arxiv.org/abs/2107.07651) employed hard negative mining at the embeddings layer using vector similarity, our method leverages LLMs.
>
> We appreciate the reviewer pointing out the concurrent work. Notably, it was published in the same month as the NeurIPS submission deadline. While that study emphasizes video-text alignment using verb perturbations, our focus is broader, encompassing transformations of objects, relationships, and attributes.
>
> Finally, our approach extends to synthetic images created by text-to-image models, while previous works focused on natural images or videos.
>
> ### Inclusion of More Examples [weakness 3]
>
> Based on your feedback, we've enriched Appendix A.2 with additional examples from (a) instances from SeeTRUE; (b) the VQ$^2$ method; and (c) the ConGen technique. This will underscore the versatility of ConGen in handling both real and synthetic images, as well as the variety of contradictions it can generate.

---

> > ### Comment · Reviewer_j32y · 2023-08-12
> > **Post rebuttal response**
> >
> > Thank you for your efforts in answering my questions.
> > After reading the authors' rebuttal, I have increased my rating to a 6 - Weak Accept

---

### Author Rebuttal · Authors · 2023-08-09

We appreciate the reviewers highlighting the comprehensiveness of our SeeTRUE benchmark and its potential contribution to the field (j32y). The noted effectiveness of our visual entailment evaluation and advancements in negative captions generation, particularly in addressing prior benchmark limitations, were recognized (prMg & xP6K). The emphasis on our significant human judgments collection (UX4d) reinforces our dedication to the text-image alignment domain.

Below, we address each reviewer's comments and questions in detail.

---

### Decision · Program_Chairs · 2023-09-21

**Decision:**

Accept (poster)

**Comment:**

This paper proposes a new benchmark for image-text alignment which provides more challenging negative examples than most prior work. To help ensure accurate labels, the authors also included human judgments that help alleviate some problems with ambiguous cases that can arise due to sparse labels.  The authors also present some results with unnatural images that a lot of prior work ignores.  The paper was evaluated by four reviewers that uniformly recommended acceptance, and the AC finds no reason to overturn their recommendation.